# Assessing Rabies Vaccine Protection against a Novel Lyssavirus, Kotalahti Bat Lyssavirus

**DOI:** 10.3390/v13050947

**Published:** 2021-05-20

**Authors:** Rebecca Shipley, Edward Wright, Fabian Z. X. Lean, David Selden, Daniel L. Horton, Anthony R. Fooks, Ashley C. Banyard

**Affiliations:** 1Wildlife Zoonoses and Vector-Borne Diseases Research Group, Animal and Plant Health Agency (APHA), Weybridge, London KT15 3NB, UK; rebecca.shipley@apha.gov.uk (R.S.); david.selden@apha.gov.uk (D.S.); tony.fooks@apha.gov.uk (A.R.F.); 2School of Life Sciences, University of Sussex, Falmer, Brighton BN1 9QG, UK; ew323@sussex.ac.uk; 3Pathology Department, Animal and Plant Health Agency (APHA), New Haw, Addlestone KT15 3NB, UK; fabian.lean@apha.gov.uk; 4Department of Pathology and Infectious Diseases, School of Veterinary Medicine, University of Surrey, Guildford GU2 7XH, UK; d.horton@surrey.ac.uk; 5Institute for Infection and Immunity, St. George’s Hospital Medical School, University of London, London SW17 0RE, UK

**Keywords:** rabies, lyssavirus, Kotalahti bat lyssavirus, KBLV, bats, vaccine protection, neutralisation, emerging, novel, zoonoses

## Abstract

Rabies is a fatal encephalitis caused by an important group of viruses within the *Lyssavirus* genus. The prototype virus, rabies virus, is still the most commonly reported lyssavirus and causes approximately 59,000 human fatalities annually. The human and animal burden of the other lyssavirus species is undefined. The original reports for the novel lyssavirus, Kotalahti bat lyssavirus (KBLV), were based on the detection of viral RNA alone. In this report we describe the successful generation of a live recombinant virus, cSN-KBLV; where the full-length genome clone of RABV vaccine strain, SAD-B19, was constructed with the glycoprotein of KBLV. Subsequent in vitro characterisation of cSN-KBLV is described here. In addition, the ability of a human rabies vaccine to confer protective immunity in vivo following challenge with this recombinant virus was assessed. Naïve or vaccinated mice were infected intracerebrally with a dose of 100 focus-forming units/30 µL of cSN-KBLV; all naïve mice and 8% (*n* = 1/12) of the vaccinated mice succumbed to the challenge, whilst 92% (*n* = 11/12) of the vaccinated mice survived to the end of the experiment. This report provides strong evidence for cross-neutralisation and cross-protection of cSN-KBLV using purified Vero cell rabies vaccine.

## 1. Introduction

Rabies is caused by viruses classified within the *Lyssavirus* genus, family Rhabdoviridae, order Mononegavirales [1]. These viruses are of importance to both human and animal health given the invariably fatal outcome from developing a neurological disease [2]. The genus *Lyssavirus* is divided into 17 species accepted by the International Committee on the Taxonomy of Viruses (ICTV) with two tentative novel lyssaviruses, Kotalahti bat lyssavirus (KBLV) and Matlo bat lyssavirus (MBLV), awaiting classification [3,4,5]. The ancestral lyssavirus reservoir host species is universally accepted as being members of the order Chiroptera with all but two lyssaviruses, Mokola and Ikoma lyssaviruses, having been detected in different bat species [6,7,8]. As well as lyssaviruses, bats have been associated as the host reservoir for an abundance of viruses, including coronaviruses (SARS and MERS viruses), filoviruses (Marburg virus and the newly discovered Měnglà virus), and Henipaviruses [9,10,11].

Within the *Lyssavirus* genus, the most broadly distributed and important to veterinary and public health is the rabies virus (RABV). Though this virus has largely been eliminated throughout western Europe, RABV is still the most commonly reported lyssavirus. RABV causes approximately 59,000 human deaths annually with the majority of these fatalities (>99%) being associated with dog-mediated transmission [12]. The human and animal burden of the other lyssaviruses, however, is largely undefined, with only 14 human lyssavirus-related deaths being reported [13,14]. This low number of reported cases may be because diagnostic capabilities often reliant on antigen detection or a clinical diagnosis, particularly in endemic areas, are unable to distinguish between different lyssavirus species [13,15].

The negative-stranded RNA lyssavirus genome consists of five genes that encode the nucleoprotein (N), phosphoprotein (P), matrix protein (M), glycoprotein (G), and viral RNA polymerase (L). Within the *Lyssavirus* genus, the species are genetically categorised according to the homology of nucleotide sequence encoding N and have been proposed to form distinct phylogroups following a combination of genetic and antigenic assessment [16,17]. Currently, rabies vaccines marketed for human use are based on whole inactivated RABV particles. It is understood that rabies vaccines confer protective immunity against phylogroup I lyssaviruses, which includes classical RABVs, Australian bat lyssavirus (ABLV), Aravan virus (ARAV), Bokeloh bat lyssavirus (BBLV), Duvenhage virus (DUVV), European bat lyssavirus type 1 (EBLV-1), European bat lyssavirus type 2 (EBLV-2), Gannoruwa bat lyssavirus (GBLV), Irkut virus (IRKV), and Khujand virus (KHUV) [18,19,20]. Certainly, studies undertaken have indicated that the level of protective immunity afforded by current vaccines against phylogroup I lyssaviruses correlate to the antibody titre level induced in the recipient of the vaccine. An antibody titre of 0.5 IU/mL is generally considered a conservative threshold above which protection from classical rabies virus strains is likely [21]. However, it has been suggested that a higher neutralising antibody titre is required to confer protection against non-RABV phylogroup I lyssaviruses [22]. Current evidence from an in vivo study suggested that an antibody titre of ≥10 IU/mL is required for protection against BBLV whilst in vitro ≥4.5 IU/mL is required for complete neutralisation of EBLV-1, EBLV-2, and ABLV [18,23]. Nevertheless, a defined cut-off for protection against non-RABV phylogroup I lyssaviruses remains to be established. Conversely, current RABV vaccines elicit no level of protective immunity against the more divergent lyssaviruses in phylogroup II, which includes Lagos bat virus (LBV Lineages A–D), Mokola virus (MOKV), and Shimoni bat virus (SHIBV), as well as highly divergent viruses tentatively assigned to a third phylogroup, which includes Ikoma lyssavirus (IKOV), Lleida bat lyssavirus (LLEBV), and West Caucasian bat virus (WCBV) [24,25,26].

Whilst RABV is the most commonly reported lyssavirus in terrestrial species, the epidemiology of non-RABV lyssaviruses is less well understood. ARAV, KHUV, SHIBV, and IKOV, specifically, exist as single detections. Consequently, gaps in epidemiological distribution, host range, and ability to cross the species barrier remain. The continual discovery of novel lyssaviruses in bats has warranted an increasing interest in the degree of vaccine cross-protection afforded by rabies vaccines [13].

In 2017, nucleic acid from a novel lyssavirus, termed Kotalahti bat lyssavirus (KBLV), was discovered in a Brandt’s bat (*Myotis brandtii*) in Kotalahti, Finland [27]. Attempts to isolate the virus were unsuccessful, likely because the bat was in a considerable stage of decomposition on arrival to the Finnish Food Safety Authority Evira [27]. Initial genetic assessment enabled amplification of a small fragment of the genome and sequencing of the nucleoprotein encoding gene demonstrated that KBLV was most closely related to KHUV (81%), followed by ARAV (79.7%), BBLV (79.5%), and EBLV-2 (79.4%), at the nucleotide level [27]. A follow-up investigation using deep sequencing successfully obtained the full KBLV genome sequence consisting of 11,878 nucleotides encoding for the five genes N, P, M, G, and L. Consequently, it was determined that KBLV is phylogenetically more closely related to EBLV-2 and clusters with KHUV and BBLV [4]. It was also determined that sera, derived from humans vaccinated against RABV, demonstrated cross-neutralising activity against KBLV G pseudotyped RABV particles [4]. Despite the preliminary findings, attempts to isolate a live replication competent KBLV virus or recombinant virus were unsuccessful. This has prevented the in vivo evaluation of vaccine efficacy as well as full antigenic characterization.

The lyssavirus glycoprotein (G) is the only viral surface protein and is the sole target for neutralising antibodies [28]. Previous studies have utilised reverse genetics (RG) techniques to exchange G proteins derived from other lyssavirus species into a RABV vaccine RG backbone to enable assessment of vaccine protection against recombinant viruses [29,30,31].

The present study describes the development of a live recombinant RABV where the RABV G has been replaced by KBLV G. In vitro and in vivo studies were used to define the ability of vaccine-derived sera and vaccine-induced antibodies (anti-RABV) to neutralise and protect against a recombinant virus containing the KBLV G protein in a murine model.

## 2. Materials and Methods

### 2.1. Cells

BHK-21 (BHK) cells were propagated in Glasgow’s modified Eagle’s medium (GMEM; Gibco, Loughborough, UK) supplemented with 10% FBS, 100 U/mL penicillin, 100 µg/mL streptomycin and 25 U/mL mycostatin (Invitrogen, Paisley, UK). BSR-T7/5 cells (BHK-derived cells that stably express T7 RNA polymerase—a kind gift from Dr Stefan Finke, FLI, Greifswald, Germany) were propagated in Dulbecco’s modified Eagle’s medium (DMEM; Gibco) supplemented with 5–10% FBS, 100 U/mL penicillin, 100 µg/mL streptomycin, and 25 U/mL mycostatin (Invitrogen).

### 2.2. Full-Length Plasmid Construction

The cDNA clone of the SN strain of RABV (cSN) was derived from the SAD B19 cDNA clone as described previously [28,32,33]. Subsequently, cSN was used as a DNA backbone vector for the assembly of the recombinant virus by exchanging cSN RABV G with KBLV G through the process of Gibson assembly (New England Biolabs (NEB), Hitchin, UK) [34]. The KBLV G open reading frame (ORF) (European Nucleotide Archive accession no. PRJEB41152) was amplified with 20 bp overhangs directed at the 5′ and 3′ end of cSN using the Q5^®^ Hot Start High-Fidelity 2X Master Mix (NEB) and the following primers: KBLVFwd (5′-CTCAAAAGACCCCGGGAAAGATGCCATTTCAAGCGGTTC-3′) and KBLVRev (5′-GTTGAAAGGCTAGCCAGTCCCTAGGCCTGAGACTGATC-3′). The underlined sequences within the primers are the incorporated overhangs complementary to cSN. To generate a linear cSN lacking the RABV G, the following primers and the same PCR master mix were used: cSNFwd (5′-GGACTGGCTAGCCTTTCAAC-3′) and cSNrev (5′-CTTTCCCGGGGTCTTTTG-3′). To confirm the presence of PCR amplicons of the appropriate size, reaction products were assessed via agarose gel electrophoresis. The insert and vector PCR products were purified using the Monarch^®^ PCR & DNA Cleanup Kit (NEB). Dpn1 (NEB) digestion was performed on the purified PCR products to remove residual vector DNA prior to recombinant genome assembly of the insert (KBLV G) and vector (cSN) DNA using the NEBuilder^®^ HiFi DNA Assembly Cloning Kit (NEB). Following transformation, plasmid propagation and plasmid harvest, plasmids were sequenced before virus rescue was attempted.

### 2.3. Virus Rescue and Titration

Virus rescue for cSN-KBLV was undertaken as described previously [30,31]. Briefly, BSR-T7/5 cells were seeded at 3 × 10^4^ cells per well in a 96-well plate (Corning, High Wycombe, UK) 18 h before transfection and 200 ng of the full-length genome plasmid and 40 ng of each of three helper plasmids, encoding the RABV N, P, and L proteins, were transfected into cells using the FuGENE^®^6 Transfection Reagent (Promega, Southampton, UK). Transfected cells were then passaged with fresh BSR-T7/5 cells at a 1:1 ratio in a 12-well plate following a 72–96 h incubation at 37 °C. Further passage was undertaken, and cells were assessed for virus antigen as described previously [35]. Where virus antigen was detected, supernatant was aspirated off and passaged further in BHK cells until 100% infection of the cell monolayer was achieved. Final stocks of cSN-KBLV were sequenced in their entirety using whole genome sequencing to confirm the expected sequence.

### 2.4. Virus Titration and Growth Kinetics

Virus titre was determined in triplicate in BHK cells as previously described [32]. Cells were seeded at 3 × 10^5^ cells per well and infected with virus serially diluted 10-fold in a 96-well plate (Corning). Cells were fixed with 80% acetone and stained with FITC anti-rabies monoclonal globulin (Fujirebio, Tokyo, Japan) at 48 h post-infection (hpi). Fluorescent foci were counted and recorded as fluorescent focus forming units per mL (ffu/mL).

Virus growth for cSN-KBLV was assessed using a multistep growth curve in comparison with the parent virus cSN as described previously [30]. Cells were then infected with each virus at a multiplicity of infection (moi) of 0.1 and at set time points (0 h, 6 h, 12 h, 24 h, 48 h, 72 h, 96 h, 120 h) 150 µL of media was removed and frozen at −80 °C until required. After the course of the experiment, each of the time point aliquots were titrated in triplicate on fresh BHK cells as three independent titration experiments. Titres were compared using the Mann–Whitney test.

### 2.5. In Vitro Studies

To assess cross-neutralisation, a modified version of the fluorescent antibody virus neutralisation test (mFAVN) was used as described previously [36,37]. Excluding cSN-KBLV, a panel of 12 lyssaviruses (Table 1) and a panel of polyclonal antisera raised against 9 viruses was used to assess cross-neutralisation. Three RABV isolates were used for comparison; a wild or street RABV isolate (designated RABV in this study), the rabies challenge standard virus-11 isolate (designated CVS in this study), and the RABV parent virus isolate used as the backbone for cSN-KBLV (designated cSN in this study). Where standardised control sera were required, the following three RABV-specific polyclonal sera were used: WHO serum, OIE serum, and hyperimmune human sera from a Human diploid cell vaccine (HDCV) (Rabies Vaccine BP; Pasteur Merieux, Lyon, France) recipient. There are no internationally agreed cut-offs for interpreting neutralising antibody titres against lyssaviruses other than CVS. As a result, for mFAVN, a threshold (reciprocal titre of 1:16) was chosen by comparison with standard sera (0.5 IU/mL) that exhibits a similar titre against CVS. 

### 2.6. In Vivo Studies

All in vivo experimentation was undertaken in ACDP3/SAPO4 biocontainment facilities at the Animal and Plant Health Agency (APHA), Weybridge, UK, and complied with strict UK Home Office regulations under the Animals in Scientific Procedures Act (1986) and Home Office license PCA17EA73. Three-to-four-week-old female BALB/c mice were purchased from a commercial supplier (Charles River, Margate, UK). Mice (*n* = 12) were vaccinated with 0.5 mL of diluted human rabies vaccine VeroRab (Novartis, Surrey, UK) via the intraperitoneal (ip) route at days 0 and 7. Mock-vaccinated mice (*n* = 8) were inoculated with MEM alone via ip. At day 29 post-vaccination, both the vaccinated and mock-vaccinated mice were challenged intracranially (ic) with 100 ffu/30 µL of cSN-KBLV. Following virus challenge, mice were observed twice daily for 22 days and clinical signs were scored using a scale of 0–5 (where 0, no effect; 1, hunched body/ruffled fur; 2, limb twitching; 3, hindquarter paralysis; 4, progressive paralysis; 5, terminal recumbency/death) [47]. The humane endpoint for termination was a clinical score of 1–2 where mice were bled under terminal anaesthesia followed by cervical dislocation. Kaplan–Meier survival curves and the log-rank Mantel–Cox test were used to analyse survivorship rates.

### 2.7. Virus Detection

A fluorescent antibody test (FAT) was undertaken as described previously [48]. Briefly, brain tissue was excised from the cerebellum and two impression smears were made on a microscope slide. Slides were left to air dry for 5 minutes at room temperature before being fixed in 99–100% acetone, washed, rinsed, and stained using FITC anti-rabies monoclonal globulin (Fujirebio, 800-092). Fluorescent green foci would indicate the presence of the N protein thereby indicating virus infection. Positive control material consisted of previously extracted CVS-infected mouse brain. Two mice from each group were tested.

Histopathology and immunohistochemistry were performed as described previously [49]. Mouse carcasses were fixed in 10% neutral buffered formalin for 2 weeks before being sectioned coronally and processed into paraffin tissue blocks. Serial sections of 4 µm were prepared and stained with haematoxylin and eosin (H&E) for histopathology or immunohistochemistry for rabies virus nucleoprotein using a monoclonal antibody, mAb 5B12 (MyBioSource Inc., San Diego, CA, USA).

### 2.8. Molecular Analyses

A real-time SYBR based reverse transcription PCR (RT-PCR) assay (Bio-Rad, Hemel Hemstead, UK) was used to determine the presence or absence of lyssavirus RNA in the mouse samples. Nucleic acids were extracted from brain tissue using Trizol (Invitrogen) according to manufacturer’s instructions. A SYBR real-time RT-PCR with N gene-specific primers and 1µg/µL of RNA was undertaken as described previously [50,51]. For semi-quantitative assessment of viral RNA, cDNA of CVS, at a known concentration (copies/g), was serially diluted from 10^−1^ to 10^−8^ in nuclease-free water and used alongside the mouse brain RNA extracts as a standard curve on the real-time RT-PCR assay. This standard curve was used to semi-quantitate the presence of viral RNA in each sample tested by comparison across Ct values. Amplification using beta actin-specific primers was used as a control for RNA extraction as described previously [51]. Two mice from each group were tested in triplicate.

### 2.9. Serology

The dorsal vein of each mouse was nicked and blood was collected in CB300 blood collection tubes with clotting activator (Sarstedt, Leicester, UK) at 21 days post-vaccination to assess seroconversion to the VeroRab vaccine. Each serum sample was tested with a standard fluorescent antibody virus neutralisation (FAVN) test as described previously [52]. At the termination of the experiment, mice were cardiac bled under terminal anesthesia and sera were assessed by FAVN against CVS and modified FAVN against cSN-KBLV. Virus-neutralising antibody titres were compared by two-way ANOVA and Tukey’s multiple comparisons tests.

### 2.10. Antigenic Cartography

Lyssavirus antigenic maps were generated from the mFAVN data as described previously [37,53]. To optimise the relative positions of the viruses and sera on a map, metric and ordinal multidimensional scaling techniques were used. Where neutralisation occurred, each virus was positioned by multiple sera and vice versa. Hence, antisera with different neutralisation profiles to the homologous viruses could be in separate locations on the map but still contribute to the positioning of the viruses. Whilst resolution increased with each increasing dimension, 3D maps were used to determine antigenic distance and visualisation as the incremental increase in precision became negligible beyond three dimensions [37]. For highlighting antigenic clusters, 2D maps were included for clarity.

### 2.11. Phylogenetics

Phylograms were constructed in MEGA6 using the neighbour-joining (NJ) algorithm after sequences were aligned. The sequences were bootstrap re-sampled 1000 times to assess the robustness of each tree node. The evolutionary distances were computed using the maximum composite likelihood method and measured in the units of the number of base substitutions per site. The resulting tree was viewed in FigTree v1.4.2 (University of Edinburgh, Edinburgh, UK; http://tree.bio.ed.ac.uk/software/figtree/ (accessed on 20 March 2021)).

## 3. Results

### 3.1. Virus Rescue and Titration

The cSN-KBLV virus was successfully rescued using RABV helper plasmids. After 5 passages, cSN-KBLV reached 100% infectivity whilst cSN only required 2 passages. Following the final passage to achieve 100% infectivity, the virus was harvested and subsequent whole genome sequencing confirmed 100% nucleotide identity to the original DNA cSN-KBLV clone (data not shown). Harvested virus was titrated in which the cSN-KBLV grew to a peak titre of 2.5 × 10^4^ ffu/mL whilst the parent cSN strain grew to a peak titre of 1.6 × 10^6^ ffu/mL. Both values, however, are comparable to similar studies using EBLV-1, EBLV-2 G, IKOV G, and WCBV G within a cSN vector [30,31].

### 3.2. Growth Kinetics

Due to the fact that the wildtype virus for KBLV had not been successfully isolated [4], the growth kinetics of cSN-KBLV were compared to that of cSN using multistep growth curves (Figure 1). By 24 hpi, both viruses were detected at around 10^3^ ffu/mL. There was no statistically significant difference between the peak viral titres observed, however, cSN grew to the highest peak titre of 7.6 × 10^7^ ffu/mL at around 96 hpi and cSN-KBLV grew to a peak titre of 1.55 × 10^7^ ffu/mL at 72 hpi. The endpoint titres showed a sigificant difference between the two viruses with cSN-KBLV exhibiting a titre of 3.33 × 10^6^ ffu/mL compared to cSN at 4.3 × 10^7^ ffu/mL (*p* < 0.001).

### 3.3. In Vitro Studies

Two experimental approaches were taken to assess cSN-KBLV antigenically: (i) to determine the titre of standard sera sufficient to neutralise cSN-KBLV over the 0.5 IU/mL value assigned as the cut-off for protection, and (ii) to evaluate for cross-neutralisation across the *lyssavirus* genus.

#### 3.3.1. Assessment of cSN-KBLV Neutralisation Using Internationally Standardised Sera

A modified FAVN was used to test the comparative neutralisation of cSN-KBLV, cSN and CVS against increasing titres of two standard sera, OIE and WHO. Both the OIE and WHO sera were capable of neutralising all three viruses at 1 IU/mL and at 2.5 IU/mL, respectively (Figure 2). Both CVS and cSN were neutralised by both OIE and WHO sera with complete neutralisation at 0.5 IU/mL. Both serological standards demonstrated the lowest level of neutralising antibodies against cSN-KBLV as 1 IU/mL of OIE sera and 2.5 IU/mL of WHO sera were required to neutralise cSN-KBLV to above the 0.5 IU/mL threshold for WHO and OIE sera against CVS. Additionally, the trend in neutralising antibody levels differed between each of the standard sera used.

OIE sera consistently showed higher levels of neutralising antibodies than WHO sera when tested against each of the viruses, though the greatest difference was observed with cSN-KBLV. At 5 IU/mL, OIE sera showed almost a 10-fold increase in neutralising antibody titre against cSN-KBLV than the WHO sera, and almost double for CVS and cSN.

#### 3.3.2. Ability of Phylogroup I-Specific Sera to Neutralise cSN-KBLV

A mFAVN was used to assess cross-neutralisation of lyssavirus-specific sera against cSN-KBLV in addition to CVS and cSN as comparison (Figure 3). Varying levels of cross-neutralisation were observed with phylogroup I lyssavirus-specific sera described in Table 1. The virus cSN was most readily neutralised by the phylogroup I sera panel with seven lyssavirus-specific sera showing levels of neutralising antibodies above 0.5 IU/mL whereas cSN-KBLV and CVS were neutralised by five lyssavirus-specific sera. For cSN-KBLV, ARAV, BBLV, EBLV-1, EBLV-2, and GBLV-specific sera exhibited a titre of neutralising antibodies above the 0.5 IU/mL cut-off whilst ABLV, DUVV, KHUV, and RABV-specific sera did not. In addition, the ability of each sera to cross-neutralise each virus varied. For cSN-KBLV, EBLV-1-specific sera showed the highest titre of neutralising antibodies. BBLV-specific sera exhibited the greatest titre of neutralising antibodies against CVS. Specific sera for a wild/street RABV strain showed the highest titre of neutralising antibodies against cSN, closely followed by BBLV and EBLV-1. Interestingly, the wild/street RABV-specific sera were only capable of neutralising cSN, indicating possible antigenic divergence of wild/street strains to the cell culture-adapted CVS used regularly in diagnostic assays. Phylogroup II-specific sera were unable to neutralise all three viruses tested (data not shown) and IRKV-specific sera were unavailable.

#### 3.3.3. Antigenic Cartography

Antigenic cartography was used to quantify antigenic relationships between phylogroup I viruses described in Table 1 and enable comparison to the evolutionary relationships facilitated via phylogenetic nucleotide analysis (Figure 4). Results showed cSN-KBLV was antigenically distinct from the other phylogroup I lyssaviruses, including the parent virus, cSN (Figure 4A,C,D). Based on the antigenic distances on the 3D map, cSN-KBLV is antigenically closest to cSN (1.00 AU), ARAV (1.21 AU), IRKV (1.45 AU), and EBLV-2 (1.65 AU). To further investigate the genetic basis of the antigenic distances, the evolutionary history of the glycoprotein nucleotide sequences of each phylogroup I lyssavirus was quantitatively inferred using the neighbour-joining method and the evolutionary distances were computed using the maximum composite likelihood (ML) method and displayed as a phylogenetic tree (Figure 4B). Based on the glycoprotein evolutionary distances, KBLV is most closely related to EBLV-2, KHUV, and BBLV. The correlation between antigenic distance and ML phylogenetic tree distance could be inferred by comparing the evolutionary and antigenic distances between cSN and each of the viruses. Based on this data, 1 AU was equal to an ML distance of 0.235 (*p* = 0.007) when RABV and CVS were excluded from the dataset. This was due to the fact that RABV G and CVS G are highly genetically similar to cSN G but are antigenically distinct, highlighting that it is not always possible to infer antigenic distance based on genetic data alone.

### 3.4. In Vivo Vaccination Challenge Study

#### 3.4.1. Vaccination and Survival

To assess the ability of vaccine-induced neutralising antibodies to protect mice from challenge with cSN-KBLV, mice were vaccinated as described and a standard FAVN test was used to determine serological response. All mice seroconverted to a titre above the internationally assigned cut-off for neutralisation of RABV, 0.5 IU/mL, with responses ranging from 0.87 IU/mL to 23.38 IU/mL 21 days after vaccination (Figure 5A). Mice were challenged via the ic route on day 29 post-vaccination with 100 ffu/30 µL of cSN-KBLV and clinical outcomes were compared to mock-vaccinated mice challenged with the same dose of cSN-KBLV. A statistically significant difference (*p* < 0.001) in survival was observed between the vaccinated and mock-vaccinated groups (Figure 5B). All mice in the mock-vaccinated group (*n* = 8) reached clinical endpoint by 7 days post-challenge, 1 day after onset of clinical disease. The first sign of clinical disease was ruffled fur. Subsequent clinical signs observed included piloerection, tail-biting, intermittent hyperactivity, and hunched stance. In the vaccinated group (*n* = 12), one mouse was terminated after developing score 2 clinical signs, which included an extreme hunched stance and limb twitching. The time from initiation of disease to termination in line with humane endpoints was protracted, however, with the mouse reaching clinical endpoint at day 13 post-challenge, 3 days after the onset of clinical disease. In addition, the vaccinated mouse that succumbed to infection had the lowest neutralising antibody titre of 0.87 IU/mL prior to challenge (indicated by ▼ on the graph). In contrast, all mice with virus-neutralising antibody titres of 1.5 IU/mL and above survived the challenge.

#### 3.4.2. Serological Responses to Infection and Post-Vaccination Challenge

All animals that succumbed to clinical disease or survived until the end of the experiment were cardiac bled before being humanely terminated. Sera were assessed for seroconversion using both the standard FAVN against CVS and modified FAVN against cSN-KBLV. In the mock-vaccinated group (*n* = 8), 63% (*n* = 5/8) of the animals had an antibody response capable of neutralising cSN-KBLV using the modified FAVN. Of the mouse sera that exhibited neutralising antibodies against cSN-KBLV, the reciprocal titres ranged from 1/16 to 1/421. In contrast, no neutralising antibody against CVS was detected with the standard FAVN (Figure 6). Despite the difference in mean reciprocal titres against CVS in the standard FAVN (1/4 ± 1/2) and cSN-KBLV in the modified FAVN (1/87 ± 1/134), the difference was not statistically significant (*p* = 0.066).

For the vaccinated group, all surviving mice developed neutralising antibody titres. In the standard FAVN against CVS, sera showed neutralising antibody titres ranging from 1/243 to 1/59049. In the modified FAVN against cSN-KBLV, sera showed less variability with neutralising antibody titres ranging from 1/729 to 1/34092. On average, however, the titre of neutralising antibodies against CVS and cSN-KBLV were not significantly different.

When comparing the serology post-challenge to that following vaccination and prior to challenge (using the standard FAVN), the titre of neutralising antibodies in the vaccinated group increased 50-fold from an average reciprocal titre of 1/284 to 1/14400 (Figure 5 and Figure 6).

Interestingly, the one mouse that succumbed to infection on day 13 post-infection showed high reciprocal titres of 1/1262 against CVS and 1/2187 against cSN-KBLV (as indicated by ▼ in both Figure 5A and Figure 6). Despite this, sera from the vaccinated group show a statically significant difference in neutralising antibody titres against CVS and cSN-KBLV than the sera derived from the mock-vaccinated group (*p* < 0.001).

#### 3.4.3. Histopathology and Immunohistochemistry

Of the mice that succumbed to cSN-KBLV, two were tested on the FAT and were positive for viral antigen. In contrast, of the mice that were vaccinated and survived challenge with cSN-KBLV, two were tested on the FAT and were all negative (data not shown). To demonstrate the pathogenicity of cSN-KBLV in the mouse model, naïve mice that succumbed to disease were evaluated by histopathology and immunohistochemistry. Infection of naïve mice with cSN-KBLV via the ic route resulted in mild to moderate, multifocal, neuronal necrosis in the cerebral cortex, thalamic (Figure 7A,B), and hippocampal regions (Figure 7C,D) of the forebrain with the association of high levels of viral antigen. Moderate to minimal amounts of virus antigen were also present multifocally within the neuroparenchyma of the thalamus, cerebellum, and brain stem, and rare antigen labelling was detected in neuronal cell bodies within the spinal cord, but histological changes were not present in these areas.

#### 3.4.4. Real-Time RT-PCR

RNA extracts from the brains of two mice from each vaccine group were evaluated via SYBR real-time RT-PCR assay (Table 2). The mouse group showing the lowest mean Ct value, 19.05 ± 0.10 and 19.37 ± 0.11, were the mice that succumbed to clinical disease in the cSN-KBLV-mock-vaccinated group, therefore indicating that this group had much higher viral RNA levels in the brain. The survivors of the vaccinated mouse group showed the highest mean Ct values with Ct values of 33.14 ± 0.53 and 36.06 ± 0.55, indicating that whilst clinical disease did not develop and FAT/IHC were negative, negligible viral RNA loads were still present in the brain at the time of termination. Based on the CVS cDNA standard curve, the cSN-KBLV-mock-vaccinated mouse group showed high copy numbers per gram (g) with 1.07 × 10^6^ copies/g and 8.53 × 10^5^ copies/g and the cSN-KBLV-vaccinated mice showed either an unquantifiable copy number or low copy number at 2.07 × 10^1^ copies/g (Figure 8).

## 4. Discussion

Rabies, caused by lyssaviruses, remains a neglected tropical disease despite being a serious threat to human and animal health globally. For lyssaviruses, the zoonotic threat from bats exists and, alongside the emergence of other bat-borne zoonoses, has been heightened due to the ongoing encroachment of humans into more remote regions/ecosystems. Though the current rabies vaccines confer protection against phylogroup I lyssaviruses, the discovery of novel lyssaviruses warrants investigation into vaccine efficacy as infection with lyssaviruses still results in a clinically incurable encephalitis.

KBLV has not been successfully isolated, nor has it been formally classified within the *lyssavirus* genus as a virus species despite being genetically related to other members of the genus. However, the availability of the KBLV genome enabled the synthesis of the recombinant virus expressing KBLV-G. In this study, the rescued virus was able to grow in both in vitro and in vivo conditions, demonstrating that the cSN backbone proteins were able to interact with the heterologous KBLV-G despite relatively low amino acid identity [4]. This further enabled in vitro cross-neutralisation assays and in vivo vaccination-challenge studies using the recombinant cSN-KBLV virus.

Growth kinetic assessment of this virus revealed that despite a successful virus rescue, endpoint titres achieved were significantly different with cSN-KBLV reaching a lower final titre compared to the parent cSN virus (Figure 1). However, the titre achieved was still higher than titres achieved in previous studies whereby more divergent glycoproteins were introduced into the same cSN backbone [31]. Decreased viral fitness of cSN-KBLV may be a result of cSN being more adapted to cell culture or a result of the parent virus (KBLV) being less able to replicate in cell culture models than the recombinant virus. This cannot be assessed as a wildtype virus isolate does not exist, although similar examples have been proposed previously. Certainly, wildtype EBLV-1 and -2 isolates grew to a lower titre than the recombinant cSN-EBLVs (referred to as cSN-1 and cSN-2), indicating that the low titres observed were not a result of the G-dependent processes such as receptor binding and viral entry but rather viral replication and assembly [30]. In a similar study, cSN expressing IKOV-G and WCBV-G have been shown to grow to viral titres around 10^5^–10^6^, almost 2 log10 lower than cSN-KBLV [31].

For the antigenic assessment of responses to viral glycoproteins in the absence of a wildtype isolate, cSN-KBLV was used. Previous studies and the data presented here reiterate that the viral G is the dominant target for neutralising antibodies and underscores the utility of recombinant vaccines for assessment where wildtype viruses are unavailable [30]. To assess the antigenicity of the KBLV-G protein, a mFAVN was undertaken with a panel of sera specific to different lyssavirus glycoproteins. Previous phylogenetic assessment determined that KBLV would be classified within phylogroup I given the nucleotide homology of 81%, 79.7%, 79.5%, and 79.4% with KHUV, ARAV, BBLV, and EBLV-2 N proteins, respectively [27]. It was concluded, following the analysis of the whole genome sequence in a second study, that KBLV is genetically most closely related to EBLV-2, KHUV, and BBLV [4]. In the neutralisation assays performed in this study, cSN-KBLV grouped within phylogroup I based on reactivity with phylogroup-specific sera with EBLV-1, BBLV, and EBLV-2-specific sera exhibited the highest titres of neutralising antibodies against cSN-KBLV (Figure 3).

Additionally, antigenic cartography was also used to quantitatively analyse antigenic data. Antigenic cartography involves geometric interpretation of binding assay data by assigning each antigen and serum a point on a 3D map such that the distance between the antigen and antiserum directly corresponds to the reciprocal titres in the cross-neutralisation assay [37]. Due to the extensive cross-neutralisation between phylogroup I species and the novel lyssavirus, cSN-KBLV, the location of these antigens on the map was fixed by multiple measurements to other antigens/antisera. Consequently, the resolution of the antigenic maps can be greater than the mFAVN assay resolution [53]. On the antigenic map, cSN-KBLV was closest to cSN, ARAV, IRKV, and EBLV-2 with antigenic distances of 1.00 AU, 1.21 AU, 1.45 AU, and 1.65 AU, respectively (Figure 4). These distances, however, must be interpreted with caution as the resolution of these antigenic maps in the average prediction error has been previously determined to be 1.22 (SE, 0.17) antigenic units in 3D [37].

Whilst EBLV-1-specific sera were most able to neutralise cSN-KBLV, being more potent than sera raised against EBLV-2, BBLV, and KHUV, this phenomenon was not visually translated in the antigenic maps with EBLV-1 showing an antigenic distance of 2.68 AU from cSN-KBLV. Interestingly, cSN was antigenically distinct from RABV and CVS; this conflicts with the findings of previous studies where RABV strains were indistinguishable from cSN and ABLV [30,37]. However, a single wildtype RABV was used in this study (RV437) and further assessment against a panel of RABVs would be needed to reliably infer antigenic differences between KBLV and circulating street RABV strains. In addition, the inclusion of additional phylogroup I-specific sera in this study may have identified distinct antigenic differences between cSN, RABV, ABLV, and CVS. CVS and RABV appear to be antigenically distant in this study and a previous study [37], indicating antigenic divergence between street RABV strains and CVS regularly used in diagnostic assays. Furthermore, the use of serological data to measure antigenic differences in cross-neutralisation assays is limited by paradoxes or irregularities in the data, such as individual variations between sera raised against the same antigen or the difficulty of absolute quantification of sera raised against different isolates for different lyssavirus species [37].

Due to the close antigenic and genetic relationship to phylogroup I, specifically EBLV-2, EBLV-1, and BBLV, it was predicted that existing rabies vaccines would be able to afford protection against the KBLV [27]. It was reported that rabies vaccines protect against BBLV in vivo with an antibody titre of ≥10 IU/mL, suggested through in vivo murine studies as being required for protection, and EBLV-1 and EBLV-2 in vitro analyses have suggested that titres ≥4.5 IU/mL are required for neutralisation [18,36]. Additionally, in a previous study investigating KBLV, sera from human vaccinees could neutralise KBLV-G pseudotyped RABV particles, however, the sera exhibited a 1.8-fold lower titre of neutralising antibodies against KBLV than against CVS [4].

In the present study, in vitro neutralisation assays demonstrated that an antibody titre of 1.0 IU/mL of OIE sera or 2.5 IU/mL of WHO sera were sufficient for neutralisation of cSN-KBLV above the serological cut-off (Figure 2). In comparison to CVS, OIE standard sera showed 1.1-fold lower neutralisation activity against cSN-KBLV. The difference in the neutralising antibody titres against cSN and cSN-KBLV confirms that G is the dominant target for neutralising antibodies. These values, however, predict the antibody titre required for protection in vivo when using a standard challenge dose of 100 TCID_50_/50 µL.

Indeed, the in vivo vaccination-challenge experiments revealed that VeroRab, a commercially available rabies vaccine, afforded almost complete protection against intracranial challenge of cSN-KBLV (Figure 5B). One vaccinated mouse succumbed to infection at day 13 but interestingly demonstrated the lowest post-vaccination antibody response as assessed via FAVN at 0.87 IU/mL (Figure 5A). Whilst above the cut-off 0.5 IU/mL value, the nature of the “severe” virus challenge through the introduction of the virus intracranially may have been too invasive to prevent productive infection in this single animal. Regardless, with this sample size we can only speculate that for protection against KBLV a higher neutralising antibody titre may be needed than that required to give complete protection against RABV in this in vivo model, in which all mice with antibody titres above 1.5 IU/mL survived ic challenge [18,36,54]. Future work with larger sample sizes would need to be performed to address this conclusively.

Following in vivo assessment, post-mortem analyses revealed that, based on two mice from each group, the vaccinated mice that survived to the end of the experiment (22 days, *n* = 11/12) were negative for lyssavirus antigen in the brain by the gold standard FAT method but very low levels of viral RNA copies could be detected in the brain using SYBR real-time RT-PCR. The lethal outcome of infection with wildtype RABV likely involves multiple mechanisms whereby viral infection drives either evasion or delay of host-immune mechanisms [55]. Whilst undefined, it is likely that the RNA detected is remnant noninfectious viral RNA from the original infection, a phenomenon reported previously in survival from RABV challenge in the murine model [56]. The mock-vaccinated mice succumbed to infection and the two mice tested in this group were positive for lyssavirus antigen both in FAT and IHC assays. Further, mice that developed clinical disease had high levels of viral RNA in the brain, and the majority remained seronegative with only very low levels of neutralising antibodies being detected (Table 2, Figure 6 and Figure 8).

Serological comparisons between the mock-vaccinated group and the vaccinated group revealed a significant difference in serological titre in both the challenge virus and vaccine virus (Figure 6). Additionally, serological comparison between the vaccinated mice post-vaccination and the same mice post-challenge revealed a 50-fold increase in neutralising antibody titre (Figure 5A and Figure 6). These findings are consistent with the hypothesis that the vaccinated mice mount a successful humoral immune response to facilitate the clearance of infection earlier than the mock-vaccinated controls, leading to low viral RNA levels in the brain by the end of the experiment [57,58]. In addition to humoral immunity, whilst poorly defined, it is likely that the cellular immune response plays an important role in viral clearance in the CNS [58,59]. Certainly, previous studies have demonstrated enhanced innate immune responses following rabies vaccination, including host responses linked with elevated BBB permeability, which enables immune cell infiltration of the CNS [59,60].

In conclusion, a recombinant cSN-KBLV virus has been generated that has been used to demonstrate the efficacy of human rabies vaccines to protect against KBLV using the recombinant virus as a surrogate both in vitro and in vivo. Whilst genetically KBLV G is most closely related to that of EBLV-2, BBLV, KHUV, and EBLV-1, cross-neutralisation assays revealed that EBLV-1, BBLV, and EBLV-2 lyssavirus-specific sera showed the highest neutralising antibodies against cSN-KBLV and antigenic map data revealed that cSN-KBLV clusters with cSN, ARAV, EBLV-2, and IRKV. Furthermore, a neutralising antibody titre of 1.5 IU/mL or above was required for the protection in a murine model. As novel lyssaviruses continue to be described, contemporary assessment of vaccine protection and evaluation of both genetic and antigenic traits is necessary for the understanding of disease mitigation strategy.

## Figures and Tables

**Figure 1 viruses-13-00947-f001:**
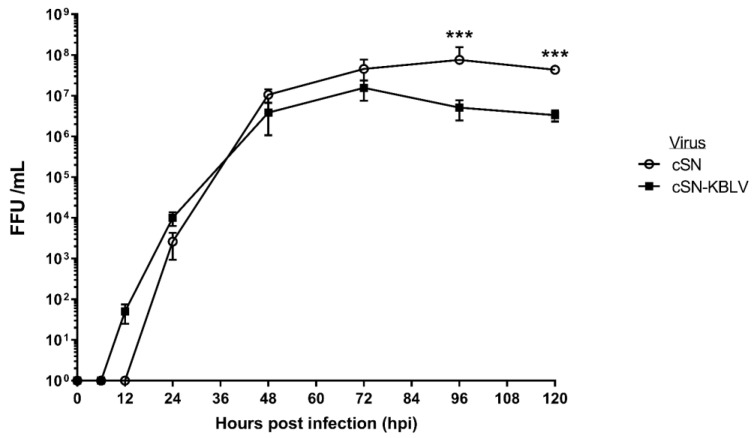
Growth kinetics of cSN-KBLV and the vaccine backbone, cSN, in vitro. For each virus, BHK-21 cells were infected with an MOI of 0.1 to produce a multiple-step growth curve over the course of 120 h. The test was performed in triplicate and the mean and standard deviation of the results plotted on a logarithmic scale. Asterisks indicate significant differences between the groups calculated using the Mann–Whitney test (*** *p* < 0.001).

**Figure 2 viruses-13-00947-f002:**
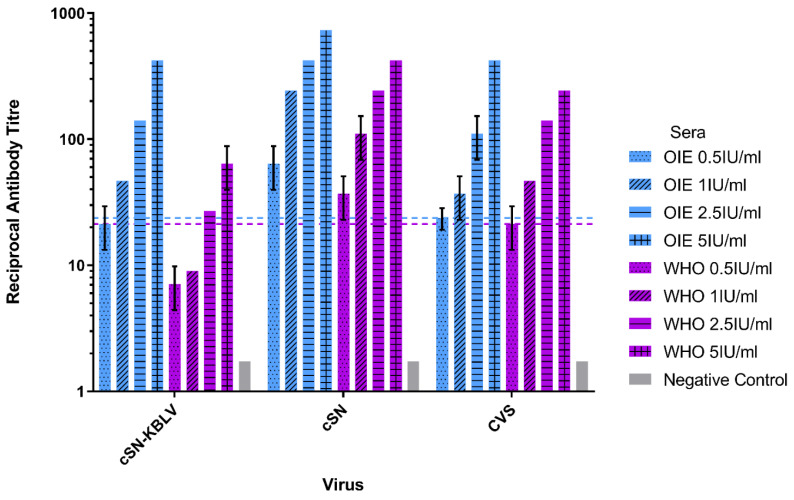
Neutralisation profiles of cSN-KBLV, cSN, and CVS against OIE and WHO standard sera using a modified fluorescent antibody virus neutralisation (mFAVN) test. The test was performed in triplicate and the mean and standard deviation of the results plotted on a logarithmic scale. The 0.5 IU/mL neutralisation cut-off is dictated by the serological standards against CVS (indicated by the coloured dashed lines—OIE = blue, WHO = purple).

**Figure 3 viruses-13-00947-f003:**
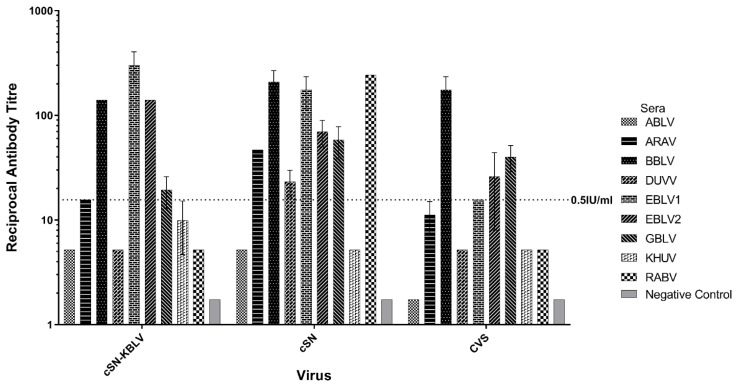
Cross-neutralisation profiles of each phylogroup I lyssavirus-specific sera using a mFAVN test. The test was performed in triplicate and the mean and standard deviation of the results plotted on a logarithmic scale. The 0.5 IU/mL neutralisation cut-off is dictated by the OIE sera against CVS (indicated by the dashed line). IRKV sera not shown.

**Figure 4 viruses-13-00947-f004:**
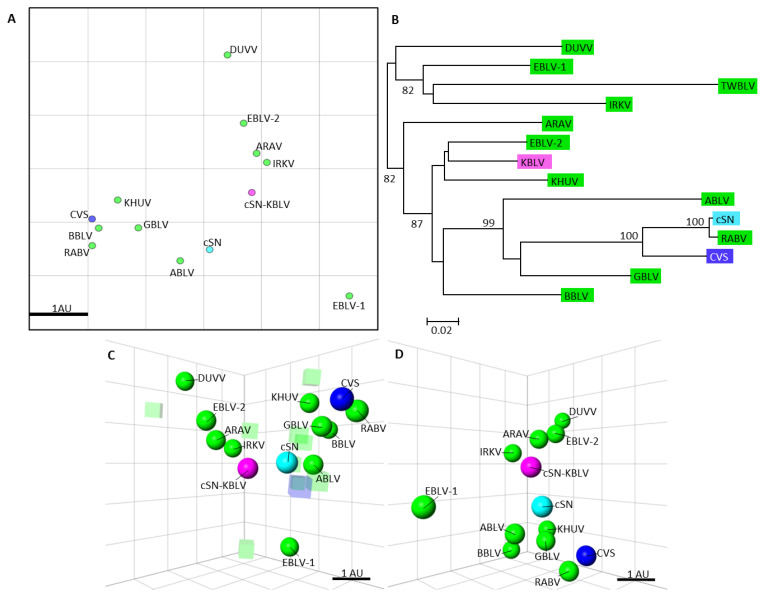
Phylogenetic tree showing the evolutionary distances of the phylogroup I lyssaviruses and antigenic cartography maps showing the antigenic distances of the phylogroup I lyssaviruses. (**A**) Two-dimensional antigenic map of Phylogroup I viruses, CVS, cSN, and cSN-KBLV. Phylogroup I lyssaviruses are coloured green, CVS coloured dark blue, cSN coloured light blue, and cSN-KBLV coloured magenta. Scale bar shows one antigenic unit with both horizontal and vertical axis representing antigenic distance so orientation within this map is free. (**B**) Phylogenetic tree of the glycoprotein nucleotide sequences. Viruses coloured as before. The evolutionary history was inferred using the neighbour-joining method and the evolutionary distances were computed using the maximum composite likelihood method and are in the units of the number of base substitutions per site. (**C**) Three-dimensional antigenic map of lyssaviruses and sera. Viruses (spheres, coloured as before) and sera (translucent-coloured boxes) are positioned such that the distance from each serum to each virus is determined by the neutralisation titre. Multidimensional scaling was used to position both sera and viruses relative to each other, so orientation of the map within the axes is free. Scale bar shows one antigenic unit (AU). (**D**) The same antigenic map rotated to a different orientation and sera removed for clarity.

**Figure 5 viruses-13-00947-f005:**
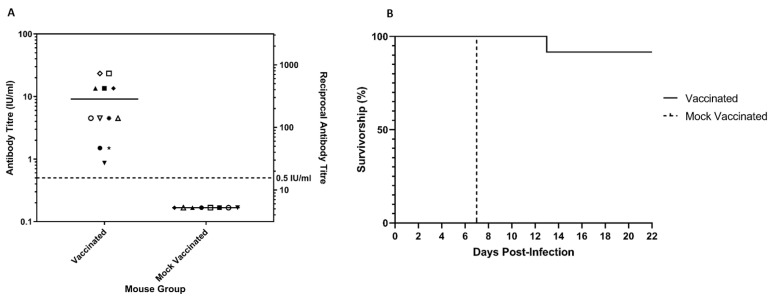
Seroconversion and survivorship of the animals. (**A**) Post-vaccination serology on day 21 for mice vaccinated with VeroRab and mice mock-vaccinated with MEM on days 0 and 7. All sera, each assigned a different symbol on the graph, were assessed for neutralising antibodies by FAVN and plotted on a logarithmic scale. The Y1 axis represents the antibody titre (IU/mL) and the Y2 axis represents the equivalent reciprocal antibody titre. (**B**) In vivo survivorship following intracranial inoculation with 100 ffu/30 µL of cSN-KBLV. Mice were vaccinated 28 days before challenge and day 0 on the graph refers to the day of challenge.

**Figure 6 viruses-13-00947-f006:**
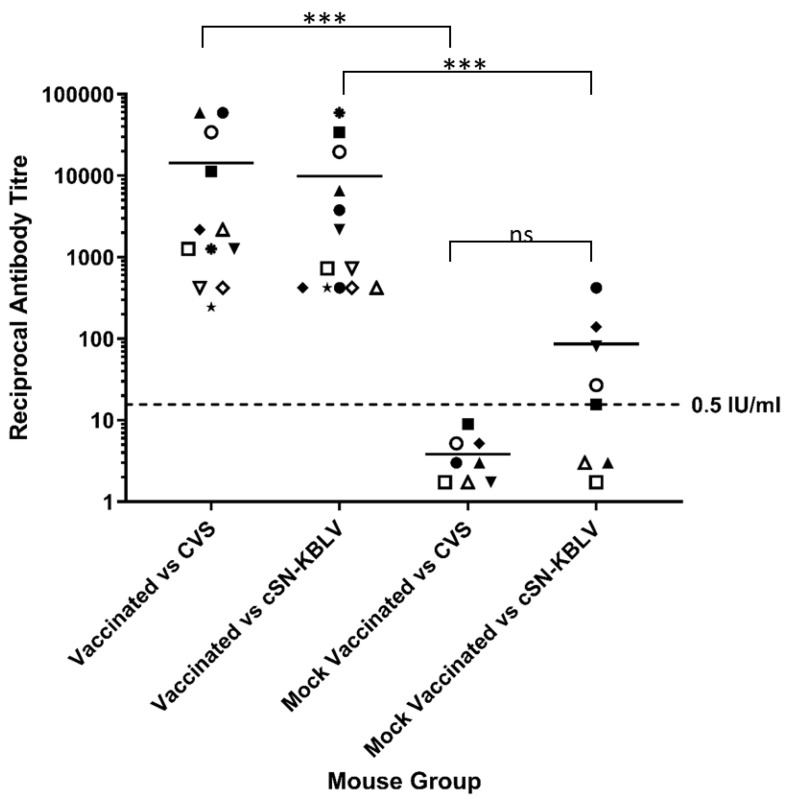
Assessment of serological status in mice following vaccination and challenge. The cSN-KBLV-challenged mice were serologically assessed against CVS and cSN-KBLV using FAVN and modified FAVN assays. Dotted line represents 0.5 IU/mL neutralisation cut-off dictated by OIE sera against CVS. Each symbol is representative of one animal. Scale is logarithmic. Asterisks indicate significant differences between the groups calculated using an ordinary two-way ANOVA and Tukey’s multiple comparisons tests (*** *p* < 0.001; ns is not significant).

**Figure 7 viruses-13-00947-f007:**
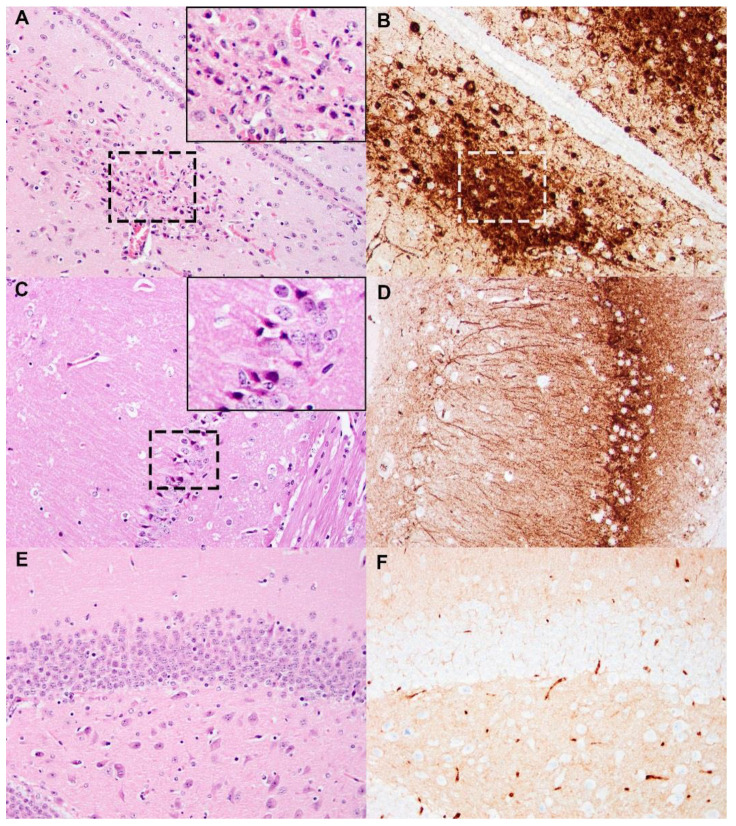
Histopathology and immunohistochemistry (IHC) against rabie nucleoprotein (N) on cSN-KBLV-infected and uninfected mouse brains. (**A**,**B**) A cSN-KBLV ic challenged mouse brain. Areas of neuronal necrosis with pyknotic nuclei and karyorrhectic cellular debris (outlined with black dashed-line box) observed in the thalamic region adjacent to the third ventricle (left, H&E stain) and serial-stained sections for N protein revealed abundant intralesional virus antigens (right, IHC, white dashed line box for region of interest co-localised with H&E section). (**C**,**D**) A cSN-KBLV IC-challenged mouse brain. Areas of neuronal necrosis characterised by red, angular and shrunken neurons (outlined with dashed-line box) in the hippocampus (left, H&E) where IHC-stained serial tissue section revealed abundant intralesional virus antigens (right, IHC). (**E**,**F**) Negative control. Representative sections showing no histological changes or virus antigens in the hippocampus. Images taken at 200x magnification. Insets show digitally magnified region of interest.

**Figure 8 viruses-13-00947-f008:**
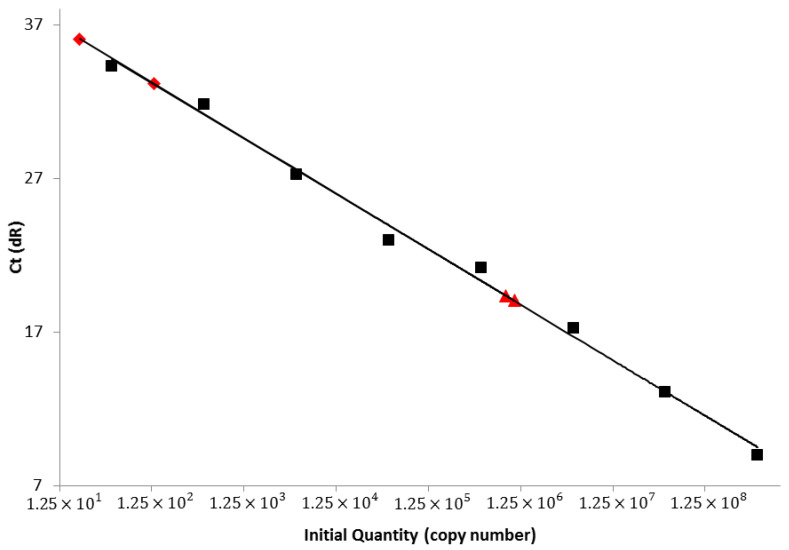
SYBR real-time RT-PCR specific to the lyssavirus nucleoprotein. Standard curve and standards are shown in black. SYBR sample values (indicated by the red symbols) plotted and interpolated using the standard curve regression line (Y = −3.607 × LOG(X) + 40.76) (*R*^2^ = 0.995; efficiency = 91%) and the mean Ct values. Triangles represent cSN-KBLV-mock-vaccinated mice and squares represent cSN-KBLV-vaccinated mice.

**Table 1 viruses-13-00947-t001:** Virus panel used and stock titre.

Designation	Lyssavirus Species	Polyclonal Antisera Used in This Study	Stock Titre (ffu/mL)	RV Number *	Isolated from	Year	Country	Genbank Accession Code ^$^	Reference
RABV	RABV	Yes	1.6 × 10^5^	RV437	Raccoon dog	-	Estonia	KF154997	[38]
CVS	RABV	No	4.3 × 10^6^	Challenge virus standard-11 strain				EU352767	[39]
cSN	RABV	No	1.2 × 10^6^	Recombinant virus; Street Alabama Dufferin (SADB19) backbone + SADB19 glycoprotein				M31046 ^^^	[40]
ABLV	ABLV	Yes	1.5 × 10^5^	RV634	Bat	1996	Australia	AY062067 (G)	[41]
ARAV	ARAV	Yes	2.0 × 10^5^	RV3379	Bat	1991	Kyrgyzstan	EF614259	[42]
BBLV	BBLV	Yes	2.5 × 10^6^	RV2507	Bat	2009	Germany	JF311903	[43]
DUVV	DUVV	Yes	3.0 × 10^6^	RV131	Bat	1986	Zimbabwe	GU936870 (G)	[37]
EBLV-1	EBLV-1	Yes	4.0 × 10^6^	RV20	Bat	1986	Denmark	KF155003	[38]
EBLV-2	EBLV-2	Yes	4.3 × 10^4^	RV628	Bat	1996	UK	KY688136	[44]
GBLV	GBLV	Yes	4.0 × 10^5^	RV3267	Bat	2015	Sri Lanka	KU244267	[45]
IRKV	IRKV	No	1.8 × 10^5^	RV3382	Bat	2002	Siberia	EF614260	[46]
KHUV	KHUV	Yes	5.0 × 10^4^	RV3380	Bat	2001	Tajikistan	EF614261	[42]

Abbreviations: ABLV, Australian bat lyssavirus; ARAV, Aravan virus; BBLV, Bokeloh bat lyssavirus; DUVV, Duvenhage virus; EBLV, European bat lyssavirus; GBLV, Gannoruwa bat lyssavirus; IRKV, Irkut virus; KHUV, Khujand virus; RABV, rabies virus. * Animal and Plant Health Agency lab identification number. ^$^ Where full genome sequence accession numbers cannot be found, glycoprotein sequence accession numbers have been included instead and are highlighted by (G). ^^^ SADB19 GenBank accession number. - Data not known.

**Table 2 viruses-13-00947-t002:** Post-mortem molecular testing on the brains of two mice from each group.

Mouse Group	SYBR Real-Time RT-PCR Assay	
	Lyssavirus Nucleoprotein	β-Actin
	Ct-Value Mean ± SD	Ct-Value Mean ± SD
cSN-KBLV-vaccinated	33.14 ± 0.53	23.23 ± 0.11
	36.06 ± 0.55	23.87 ± 0.12
cSN-KBLV-mock-vaccinated	19.37 ± 0.11	21.93 ± 0.08
	19.03 ± 0.10	21.82 ± 0.14

## Data Availability

Not applicable.

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
