# Peer review of "Assessing Rabies Vaccine Protection against a Novel Lyssavirus, Kotalahti Bat Lyssavirus"

_viruses, 2021, doi:10.3390/v13050947_

Round 1
Reviewer 1 Report
Shipley et al., Assessing rabies vaccine protection against a novel lyssavirus, Kotalahti Bat Lyssavirus.
In this manuscript, Shipley et al describe a new lyssavirus recombinant, consisting of the rabies vaccine strain SAD-B19 (cSN), in which G has been replaced with G from Kotolahti Bat Lyssavirus (KBLV). The resulting recombinant, cSN-KBLV, was shown to be replication competent and capable of producing a modest-titer stock that was subjected to both in vitro and in vivo characterization. Data show close antigenic relationship to other type I lyssavirus strains. Further, intracranial injection of cSN-KBLV resulted in lethality in unvaccinated mice, but 92% survival of mice vaccinated with VeroRab. Thus, the data suggest VeroRab can protect against KBLV, with the caveat that required protective titer is greater than the 0.5 IU/mL standard for protection against RABV infection. Overall, the data in the manuscript are of high quality and the sum of the findings is of value, particularly considering that there has been no successful rescue of KBLV to date, and there has been no previous published characterization of KBLV G in vitro cross-neutralization and in vivo neutralization. However, a number of points should be addressed to further improve this manuscript:
Materials and Methods:
- Lines 132-3 references underlined sequences within the primer sequences; however, there is no underlining
- Line 160: anti-N mAb. This reagent is insufficiently identified. Please name the clone or provide some other clear identifier (e.g., catalog number), if the clone number is not specified.
- Line 172: An appendix is included in the manuscript, which contains Table A1. In the Discussion, this table is referred to as Table 1. This table contains important information that should not be relegated to an appendix. Please rename to Table 1, and please place additional callouts for this Table in the Results text, when the different lyssavirus strains are introduced in conjunction with the various figures.
- Line 239: text refers to “Horton et al . . .” – please replace with [37]
Results
- Section starting at line 296: This section is very short, lacking in detail, and not organized in an effective manner. In particular, some emphasis should be given to a complete description of sera that do and do not neutralize cSN-KBLV, followed by contrasting to the results with cSN and CVS. This is how the actual figure is organized (i.e., cSN-KBLV is first), so it is rather confusing why the text section begins with a discussion of the data for cSN.
- Section beginning at line 315: The street strain of rabies used in this study (RV437) seems like an antigenic oddity, in contrast to ref 37, in which it clustered with all other wild RABV strains. Specifically, in the ref 37 study, all of the RABV strains were closely clustered together, with ABLV. In this study, ABLV and cSN are closely clustered, but the RV437 and and CVS are (relatively) distant by antigenic mapping. In ref 37, CVS was found to be antigenically distant (again, relatively) from all of the other RABV strains (presumably including RV437 based on the text, although this is impossible to confirm from the minimalistic labeling of the figures), which were closely clustered. While the RABV street strain (RV437) is not the focus of this study, a significant generalized conclusion by a reader (inferred from the data) might be that KBLV is antigenically distant from street strains of RABV. However, the mapped location of this street strain is in contrast to what was presented in ref 37. Although there is currently discussion of the contrasting mapping results between this study and those in refs 30 and 37, some additional text, in that same paragraph, focused on RV437, in the context of the above critique, would be helpful.
- Figures 5 and 6: In the results describing figures 5A and 6, the differing y-axes are quite frustrating. Claims in the text, such as, "When comparing the serology post-challenge . . . the titre of neutralising antibodies in the vaccinated group increased 50-fold from an average reciprocal titre of 1/284 to 1/14400 (Figure 5 and Figure 6)," really cannot be evaluated, based on the fact that the y-axes on these figures use distinct scales. I would suggest making another panel in Fig 6 where these data are directly compared, using the reciprocal antibody titer scale. Similarly, the authors have gone to the trouble of assigning an individual symbol to each animal in 5A and 6. However, there is no key so that individual animals can be identified and compared across figures, and this labeling is not carried over in to 5B. This is important, given the statement in the text, "Interestingly, the one mouse that succumbed to infection on day 13 post-infection showed high reciprocal titres of 1/1262 against CVS and 1/2187 against cSN-KBLV." The reader should be able to look at 5A, 5B and 6 and the Results text to unequivocally determine which mouse this is, in order to see how this animal compares to the rest of this group and to the other groups. Currently, the reader can only make an educated guess, based on other information in the Results text.
- Section beginning at line 400: In the histopathology section, there should be a specific description of the data for the vaccinated mouse that succumbed to infection. If those data differ from the non-vaccinated animals, the data should be included in Fig 7. Also, the text for the histopathology section has callouts for 7A (hypothalamus), 7C (hippocampus), and 7B and D (forebrain). The only text in the description of the figure from Results that matches the actual Figure 7 and legend in the manuscript is 7A. Please either replace the figure with the correct figure or replace the text with the correct text.
- Section starting at line 418: In the RT-PCR section, the meaning of "Two mouse brain RNA extracts . . ." is not clear. I presume, based on the figure, that the intended meaning is "RNA extracts from two mice . . ." Please clarify. Also, if this is the meaning, why were only two mice per group evaluated using this relatively straightforward assay? There seems to be no good rationale for not performing this analysis using RNA from all of the mice in each of these groups, and then performing statistical evaluation of the resulting data.
- Line 420: Callout for Table 1 refers to data actually present in Table 2
Discussion:
- The Discussion could be improved by adding text regarding the following:
- Possible role of cellular immunity in the observed responses to the vaccine, given that the data show a correlation with antibody titers but do not specifically demonstrate mechanistic involvement (There is evidence of the employed vaccine inducing such responses. e.g., PMID: 31358408 and references therein). In the context of this inoculation in the brain, is circulating antibody or a cellular immune response more likely to mediate control of the CNS infection? Please discuss.
- Remaining low-level virus in the brains of surviving vaccinated-infected animals. i.e., the data show control of viral infection, but not clearance. Thoughts about why the infection was not cleared would be of value.

Author Response
We would like to thank reviewer one for their constructive comments that have improved the overall quality of the manuscript.
A rebuttal to each point is presented below:
Reviewer 1:
In this manuscript, Shipley et al describe a new lyssavirus recombinant, consisting of the rabies vaccine strain SAD-B19 (cSN), in which G has been replaced with G from Kotolahti Bat Lyssavirus (KBLV). The resulting recombinant, cSN-KBLV, was shown to be replication competent and capable of producing a modest-titer stock that was subjected to both in vitro and in vivo characterization. Data show close antigenic relationship to other type I lyssavirus strains. Further, intracranial injection of cSN-KBLV resulted in lethality in unvaccinated mice, but 92% survival of mice vaccinated with VeroRab. Thus, the data suggest VeroRab can protect against KBLV, with the caveat that required protective titer is greater than the 0.5 IU/mL standard for protection against RABV infection. Overall, the data in the manuscript are of high quality and the sum of the findings is of value, particularly considering that there has been no successful rescue of KBLV to date, and there has been no previous published characterization of KBLV G in vitro cross-neutralization and in vivo neutralization. However, a number of points should be addressed to further improve this manuscript:
Materials and Methods:
- Lines 132-3 references underlined sequences within the primer sequences; however, there is no underlining
We have rectified this in the revised version
- Line 160: anti-N mAb. This reagent is insufficiently identified. Please name the clone or provide some other clear identifier (e.g., catalog number), if the clone number is not specified.
We have rectified this in the revised version
- Line 172: An appendix is included in the manuscript, which contains Table A1. In the Discussion, this table is referred to as Table 1. This table contains important information that should not be relegated to an appendix. Please rename to Table 1, and please place additional callouts for this Table in the Results text, when the different lyssavirus strains are introduced in conjunction with the various figures.
We have rectified this in the revised version
- Line 239: text refers to “Horton et al . . .” – please replace with [37]
We have rectified this in the revised version
Results
- Section starting at line 296: This section is very short, lacking in detail, and not organized in an effective manner. In particular, some emphasis should be given to a complete description of sera that do and do not neutralize cSN-KBLV, followed by contrasting to the results with cSN and CVS. This is how the actual figure is organized (i.e., cSN-KBLV is first), so it is rather confusing why the text section begins with a discussion of the data for cSN.
We have rectified this in the revised version
- Section beginning at line 315: The street strain of rabies used in this study (RV437) seems like an antigenic oddity, in contrast to ref 37, in which it clustered with all other wild RABV strains. Specifically, in the ref 37 study, all of the RABV strains were closely clustered together, with ABLV. In this study, ABLV and cSN are closely clustered, but the RV437 and and CVS are (relatively) distant by antigenic mapping. In ref 37, CVS was found to be antigenically distant (again, relatively) from all of the other RABV strains (presumably including RV437 based on the text, although this is impossible to confirm from the minimalistic labeling of the figures), which were closely clustered. While the RABV street strain (RV437) is not the focus of this study, a significant generalized conclusion by a reader (inferred from the data) might be that KBLV is antigenically distant from street strains of RABV. However, the mapped location of this street strain is in contrast to what was presented in ref 37. Although there is currently discussion of the contrasting mapping results between this study and those in refs 30 and 37, some additional text, in that same paragraph, focused on RV437, in the context of the above critique, would be helpful.
We have rectified this in the revised version
- Figures 5 and 6: In the results describing figures 5A and 6, the differing y-axes are quite frustrating. Claims in the text, such as, "When comparing the serology post-challenge . . . the titre of neutralising antibodies in the vaccinated group increased 50-fold from an average reciprocal titre of 1/284 to 1/14400 (Figure 5 and Figure 6)," really cannot be evaluated, based on the fact that the y-axes on these figures use distinct scales. I would suggest making another panel in Fig 6 where these data are directly compared, using the reciprocal antibody titer scale. Similarly, the authors have gone to the trouble of assigning an individual symbol to each animal in 5A and 6. However, there is no key so that individual animals can be identified and compared across figures, and this labeling is not carried over in to 5B. This is important, given the statement in the text, "Interestingly, the one mouse that succumbed to infection on day 13 post-infection showed high reciprocal titres of 1/1262 against CVS and 1/2187 against cSN-KBLV." The reader should be able to look at 5A, 5B and 6 and the Results text to unequivocally determine which mouse this is, in order to see how this animal compares to the rest of this group and to the other groups. Currently, the reader can only make an educated guess, based on other information in the Results text.
We have rectified this in the revised version. Figure 5A now has two Y axis, one representing antibody titre and one representing equivalent reciprocal antibody titre. In the text, we have highlighted which symbol represents the vaccinated mouse that succumbed to infection.
- Section beginning at line 400: In the histopathology section, there should be a specific description of the data for the vaccinated mouse that succumbed to infection. If those data differ from the non-vaccinated animals, the data should be included in Fig 7. Also, the text for the histopathology section has callouts for 7A (hypothalamus), 7C (hippocampus), and 7B and D (forebrain). The only text in the description of the figure from Results that matches the actual Figure 7 and legend in the manuscript is 7A. Please either replace the figure with the correct figure or replace the text with the correct text.
All mice that succumbed to infection were positive on the gold standard FAT test indicating the presence of viral antigen in the brain in both the mock vaccinated mice and the vaccinated mouse that succumbed to infection. For the histopathology and immunohistochemistry, the brain of the vaccinated mouse was not tested as the aim of histopathological examination was to determine the pathology of cSN-KBLV, rather than to determine whether the brain was positive for antigen/neuronal necrosis. We have amended the text for the second point made in this comment.
- Section starting at line 418: In the RT-PCR section, the meaning of "Two mouse brain RNA extracts . . ." is not clear. I presume, based on the figure, that the intended meaning is "RNA extracts from two mice . . ." Please clarify. Also, if this is the meaning, why were only two mice per group evaluated using this relatively straightforward assay? There seems to be no good rationale for not performing this analysis using RNA from all of the mice in each of these groups, and then performing statistical evaluation of the resulting data.
We have rectified this in the revised version. The reviewer makes a valid comment however due to the pandemic, access to site and time in containment were limited so two mice were tested as representatives of each mouse group.
- Line 420: Callout for Table 1 refers to data actually present in Table 2
We have rectified this in the revised version
Discussion:
- The Discussion could be improved by adding text regarding the following:
- Possible role of cellular immunity in the observed responses to the vaccine, given that the data show a correlation with antibody titers but do not specifically demonstrate mechanistic involvement (There is evidence of the employed vaccine inducing such responses. e.g., PMID: 31358408 and references therein). In the context of this inoculation in the brain, is circulating antibody or a cellular immune response more likely to mediate control of the CNS infection? Please discuss.
We have rectified this in the revised version
- Remaining low-level virus in the brains of surviving vaccinated-infected animals. i.e., the data show control of viral infection, but not clearance. Thoughts about why the infection was not cleared would be of value.
We think that residual RNA detected in the brain of survivors is either from abortive infection or following viral clearance. The mechanism of RNA maintenance in brain material in not understood, however other reports have published similar observations. We have amended the text to highlight this.
Reviewer 2 Report
The paper of Shipley et al describes generation of chimeric rabies virus pseudotyped with glycoprotein of the recently described Kotalahti bat lyssavirus (which was never isolated, only detected genetically); assesses serologic cross-reactivity of this construct with other lyssaviruses, and the ability of commercially available rabies vaccine to elicit protection against this virus.
This is a well designed standard study which need to be performed for every new virus with zoonotic potential. I do not have any major concerns, only several minor which are listed below as they appear in the text.
- Lines 48-49: Authors are confused between a virus, a viral species, and a prototype viral species. Based on the present complexity, ICTV removed prototype species from viral taxonomy. Otherwise, virus is the real biologic entity, named rabies virus (RABV). In contrast, Rabies lyssavirus (capitalized, latinized, and not abbreviated) is a taxonomic “container” where rabies virus is placed. In this sentence authors may simply say “…the most broadly distributed and important for veterinary and public health is rabies virus (RABV)”. They may add, if they wish, “…which belongs to the species Rabies lyssavirus” but this is not necessary, particularly as we may expect further changes in the binomial species nomenclature (which will not affect virus names, only species).
- Line 51: 59,000 human deaths (and remove “human” at the end of this sentence).
- Line 53: a species may not have any burden as it is the artificial taxonomic “container”. The real viruses do. Therefore, “other lyssaviruses”.
- Line 56: the same confusion: must be either “between different lyssaviruses” or “between lyssaviruses from different species”.
- Lines 59060: “the species are categorized genetically into species…” is not the best verbal construction.
- Lines 60-61: these are not the only demarcation criteria for lyssavirus species and phylogroups. Moreover, phylogroups have never be established officially as a part of viral taxonomy.
- Lines 65-67: Aravan virus, European bat lyssaviruses, type 1 and 2, Duvenhage virus, Irkut virus, Khujand virus.
- Lines 80-82: Mokola virus, Shimoni bat virus, West Caucasian bat virus; Phylogroup III does not exist, was never defined, as demarcation proposed for Phylogroups I and II does not work for IKOV, WCBV and LLEBV.
- Line 90: Myotis brandtii.
- Line 100: the abbreviation RABV is sufficient, no need to repeat “rabies virus”.
- Lines 131-143: there are no underlined nucleotides in the sequence as it is present in PDF file. However, I think the sequences and too detailed description are non-essential as authors did the standard Gibson assembly with 20 bp overhangs. Some shortened description with a reference will be sufficient. What is interesting for me that authors used Dpn1 during Gibson assembly “to remove residual vector DNA”. Although we used to use Dpn1 during site-directed mutagenesis, it is not a part of Gibson assembly protocol where all genetic parts are reacting (linearized backbone and the insert with overhangs) and there is no “residual vector DNA”. Pwerhaps authors should double-check their description of the procedure and, as I said above, shorten it with a reference.
- Lines 176-177: I did not understand the last sentence in this paragraph.
- Lines 1856-187: it appears that you did intracranial inoculation of ~2-month-old mice. As this is a non-routine procedure, I suggest to describe anesthesia and inoculation techniques in more details.
- Lines 210-220: which gene was targeted by qRT-PCR?
- Lines 216-217. I assume the efficiency of amplification based on the standard curve was 94%?
- Line 218: sounds like the qRT-PCR tests were not normalized either by the housekeeping gene (which was used to control RNA extraction only), or at least by the total amount of RNA taken in the RT reaction. In such settings, even “relative” copy number based on the standard curve from cDNA will not show “copy number/g”.
- Line 274: what means “the level of standard control sera”? I would appreciate if authors re-word first part of this sentence (under “i)”, lines 273-275) to make it clear.
- Line 279: “vaccine-derived standard sera” does not sound correct. These are standard sera/immune globulins of known neutralizing activity obtained after vaccination of animals with commercially available rabies vaccines.
- Lines 280-285: Based on Figure 2, I would say that all concentrations of OIE standard neutralized CVS and cSN-KBLV almost equally. Please, double check.
- Line 299: I guess “were observed”.
- Lines 302-305: what means “the greatest neutralising antibodies” and “the highest neutralizing antibodies”?
- Figure 3. I disagree with design of this figure. The 0.5IU/ml line is crossing all viruses although it is relevant to RABV (and anti-RABV antibodies) only. Furthermore, I do not think that such graphical representation is justified and demonstrative. Perhaps a table that shows neutralising titre differences between homologous and heterologous viruses would be more and easy to follow.
- Figure 4. Interesting distribution. First, no visual congruence with phylogenetic relationships (e.g., DUVV, IRKV and EBLV-1 should be close to each other but on the antigenic “map” they are placed quite distantly; the same about EBLV-2, KHUV and KBLV); second, it does not correspond to what you said in line 305, that “EBLV-1-specific sera showed the highest neutralizing antibodies” to cSN-KBLV. In any of the three projections of your 3D antigenic map I see EBLV-1 quite distant from cSN-KBLV. I understand that the 3D map should reflects reciprocal relationships, but they are hard to interpret.
- Lines 496-497: I disagree that resolution of antigenic maps is greater than resolution of FAVN data. It is just different representation of the same data. And as I mentioned above, whilst cross-neutralization data (from FAVN) relatively well correlates with phylogenetic data, implementation of these in 3D antigenic map looks somewhat awkward, has no corroboration by other methods, and the utility of such 3D map from my prospective is quite questionable. I think your following paragraph (lines 503-514) supports this my opinion, and I agree with you on “paradoxes or irregularities” (lines 511-512).
- Line 521: what is “human vaccine sera”?
- Lines 523-525: I do not understand the 1 st sentence of this paragraph. Please, consider re-wording.
- Lines 525-527. I do not understand the following sentences either. What is “OIE vaccine sera”? Figure 2 and the relevant part of Results (lines 283-290) clearly demonstrate that OIE standard had greater neutralizing activity than WHO standard, why you say here that “OIE vaccine sera showed 1.1 fold lower neutralization activity against cSN-KBLV”, what I am missing?
- Line 538: I disagree that you can conclude anything based on 1 dead animal. It might be vaccinated wrong (e.g. penetration of abdominal organs) and that might be not only the reason of the limited antibody response but general poor health of this mouse, perhaps with suppressed immune functions at different levels; or the same/similar conditions could develop irrelevant to the vaccination performed.
- Lines 544-545: do you think the low-positive SYBR PCR results reflected the remnant non-infectious viral RNA from the challenge, or it was low-level of virus replication, or it was false-positive result? A short discussion on this would be very interesting.
Author Response
We would like to thank reviewer one for their constructive comments that have improved the overall quality of the manuscript.
A rebuttal to each point is presented below:
The paper of Shipley et al describes generation of chimeric rabies virus pseudotyped with glycoprotein of the recently described Kotalahti bat lyssavirus (which was never isolated, only detected genetically); assesses serologic cross-reactivity of this construct with other lyssaviruses, and the ability of commercially available rabies vaccine to elicit protection against this virus.
This is a well designed standard study which need to be performed for every new virus with zoonotic potential. I do not have any major concerns, only several minor which are listed below as they appear in the text.
- Lines 48-49: Authors are confused between a virus, a viral species, and a prototype viral species. Based on the present complexity, ICTV removed prototype species from viral taxonomy. Otherwise, virus is the real biologic entity, named rabies virus (RABV). In contrast, Rabies lyssavirus (capitalized, latinized, and not abbreviated) is a taxonomic “container” where rabies virus is placed. In this sentence authors may simply say “…the most broadly distributed and important for veterinary and public health is rabies virus (RABV)”. They may add, if they wish, “…which belongs to the species Rabies lyssavirus” but this is not necessary, particularly as we may expect further changes in the binomial species nomenclature (which will not affect virus names, only species).
We have rectified this in the revised version
- Line 51: 59,000 human deaths (and remove “human” at the end of this sentence).
We have rectified this in the revised version
- Line 53: a species may not have any burden as it is the artificial taxonomic “container”. The real viruses do. Therefore, “other lyssaviruses”.
We have rectified this in the revised version
- Line 56: the same confusion: must be either “between different lyssaviruses” or “between lyssaviruses from different species”.
We have rectified this in the revised version
- Lines 59060: “the species are categorized genetically into species…” is not the best verbal construction.
We have rectified this in the revised version
- Lines 60-61: these are not the only demarcation criteria for lyssavirus species and phylogroups.
Moreover, phylogroups have never be established officially as a part of viral taxonomy.
We have rectified this in the revised version. Whilst assigning lyssaviruses to phylogroups is not recognized as part of viral taxonomy, based on previous publications referring to vaccine protection, genetic analyses, and phylogroups, the authors believe it important to mention phylogroups and associated vaccine protection as a way to fully characterize KBLV.
- Lines 65-67: Aravan virus, European bat lyssaviruses, type 1 and 2, Duvenhage virus, Irkut virus, Khujand virus.
We have rectified this in the revised version
- Lines 80-82: Mokola virus, Shimoni bat virus, West Caucasian bat virus; Phylogroup III does not exist, was never defined, as demarcation proposed for Phylogroups I and II does not work for IKOV, WCBV and LLEBV.
We have changed the text however reference to the existence of a third phylogroup has been explored in the literature referring to the highly divergent lyssaviruses IKOV, LLEBV, and WCBV. [1-4]
- Line 90: Myotis brandtii.
We have rectified this in the revised version
- Line 100: the abbreviation RABV is sufficient, no need to repeat “rabies virus”.
We have rectified this in the revised version
- Lines 131-143: there are no underlined nucleotides in the sequence as it is present in PDF file. However, I think the sequences and too detailed description are non-essential as authors did the standard Gibson assembly with 20 bp overhangs. Some shortened description with a reference will be sufficient. What is interesting for me that authors used Dpn1 during Gibson assembly “to remove residual vector DNA”. Although we used to use Dpn1 during site-directed mutagenesis, it is not a part of Gibson assembly protocol where all genetic parts are reacting (linearized backbone and the insert with overhangs) and there is no “residual vector DNA”. Perhaps authors should double-check their description of the procedure and, as I said above, shorten it with a reference.
We have now underlined the nucleotides in the primer sequence. Dpn1 digestion was not used during Gibson assembly but prior to assembly. The NEB protocol (https://www.neb.com/-/media/nebus/files/manuals/manuale5510.pdf) recommends digesting the PCR product(s) with Dpn1 to destroy plasmid template prior to setting up the Gibson assembly reaction.
- Lines 176-177: I did not understand the last sentence in this paragraph.
We have rectified this in the revised version. There is no internationally assigned cut-off for neutralisation when using the mFAVN so comparisons have to be drawn from the titres of standard sera against challenge virus standard used in the FAVN.
- Lines 1856-187: it appears that you did intracranial inoculation of ~2-month-old mice. As this is a non-routine procedure, I suggest to describe anesthesia and inoculation techniques in more details.
The authors believe that this is a well-established technique in vaccination-challenge experiments for rabies and other lyssaviruses [1, 2, 5-7]. Typically 3-4 week old mice are vaccinated on days 0 and 7. On day 21 post vaccination, blood is collected to assess seroconversion and day 28/29, mice are challenged IC with virus.
- Lines 210-220: which gene was targeted by qRT-PCR?
We have rectified this in the revised version- the N gene.
- Lines 216-217. I assume the efficiency of amplification based on the standard curve was 94%?
Correct
- Line 218: sounds like the qRT-PCR tests were not normalized either by the housekeeping gene (which was used to control RNA extraction only), or at least by the total amount of RNA taken in the RT reaction. In such settings, even “relative” copy number based on the standard curve from cDNA will not show “copy number/g”.
We have altered the manuscript to temper the statement and appreciate that the outputs are semi-quantitative alone.
- Line 274: what means “the level of standard control sera”? I would appreciate if authors re-word first part of this sentence (under “i)”, lines 273-275) to make it clear.
We have rectified this in the revised version
- Line 279: “vaccine-derived standard sera” does not sound correct. These are standard sera/immune globulins of known neutralizing activity obtained after vaccination of animals with commercially available rabies vaccines.
We have rectified this in the revised version
- Lines 280-285: Based on Figure 2, I would say that all concentrations of OIE standard neutralized CVS and cSN-KBLV almost equally. Please, double check.
We have reworded this section in the revised version
- Line 299: I guess “were observed”.
We have rectified this in the revised version
- Lines 302-305: what means “the greatest neutralising antibodies” and “the highest neutralizing antibodies”?
We have rectified this in the revised version
- Figure 3. I disagree with design of this figure. The 0.5IU/ml line is crossing all viruses although it is relevant to RABV (and anti-RABV antibodies) only. Furthermore, I do not think that such graphical representation is justified and demonstrative. Perhaps a table that shows neutralising titre differences between homologous and heterologous viruses would be more and easy to follow.
There is no internationally assigned cut-off for neutralisation when using the mFAVN so comparisons have to be drawn from the titres of standard sera against challenge virus standard used in the FAVN. The main focus of this figure is to draw comparisons between the backbone virus (cSN), the challenge virus standard (CVS), and cSN-KBLV. We believe that the vastly different neutralisation profiles between the three viruses and phylogroup I specific sera portrays this effectively.
- Figure 4. Interesting distribution. First, no visual congruence with phylogenetic relationships (e.g., DUVV, IRKV and EBLV-1 should be close to each other but on the antigenic “map” they are placed quite distantly; the same about EBLV-2, KHUV and KBLV); second, it does not correspond to what you said in line 305, that “EBLV-1-specific sera showed the highest neutralizing antibodies” to cSN-KBLV. In any of the three projections of your 3D antigenic map I see EBLV-1 quite distant from cSN-KBLV. I understand that the 3D map should reflects reciprocal relationships, but they are hard to interpret.
We have amended the wording for this figure to make it clearer. To clarify, antigenic maps only display data based on reciprocal antibody titre and no genetic analyses are involved in the placement of the viruses on the map. Therefore, phylogenetic relationships do not always correspond the antigenic relationships as shown for EBLV-1 and EBLV-2 in a previous publication [8]. The viruses will appear in specific positions on the map in relation to all other viruses and sera. The lyssavirus-specific serum and homologous virus can appear in two separate positions on the map in cases where the serum is broadly cross-neutralising but the homologous virus is not neutralised by other lyssavirus-specific sera (e.g. EBLV-1). Previously it has been showed that murine anti-Aravan virus (anti-ARAV) serum neutralized Khujand virus (KHUV) and ARAV equally but an anti-KHUV serum was less effective at neutralizing ARAV than KHUV [8]. Additionally, as the viruses are positioned in relation to all other viruses and sera, and not just cSN-KBLV, additional constraints are applied and the antigenic data seen in figure 3, may not always be reflected in the antigenic map of all phylogroup I viruses vs all phylogroup I-specific sera. The following citation has more information on this [9].
- Lines 496-497: I disagree that resolution of antigenic maps is greater than resolution of FAVN data. It is just different representation of the same data. And as I mentioned above, whilst cross-neutralization data (from FAVN) relatively well correlates with phylogenetic data, implementation of these in 3D antigenic map looks somewhat awkward, has no corroboration by other methods, and the utility of such 3D map from my prospective is quite questionable. I think your following paragraph (lines 503-514) supports this my opinion, and I agree with you on “paradoxes or irregularities” (lines 511-512).
The use of antigenic cartography maps allows quantitative interpretation of the serological data. The serological data alone is limited by irregularities of the modified FAVN test, such as higher heterologous than homologous titres and individual variations between sera. Because the data is plotted in relation to all viruses and all sera, you overcome these limitations [9-11].
- Line 521: what is “human vaccine sera”?
We have rectified this in the revised version
- Lines 523-525: I do not understand the 1 st sentence of this paragraph. Please, consider re-wording.
We have rectified this in the revised version
- Lines 525-527. I do not understand the following sentences either. What is “OIE vaccine sera”? Figure 2 and the relevant part of Results (lines 283-290) clearly demonstrate that OIE standard had greater neutralizing activity than WHO standard, why you say here that “OIE vaccine sera showed 1.1 fold lower neutralization activity against cSN-KBLV”, what I am missing?
We have rectified this in the revised version
- Line 538: I disagree that you can conclude anything based on 1 dead animal. It might be vaccinated wrong (e.g. penetration of abdominal organs) and that might be not only the reason of the limited antibody response but general poor health of this mouse, perhaps with suppressed immune functions at different levels; or the same/similar conditions could develop irrelevant to the vaccination performed.
We have rectified this in the revised version
- Lines 544-545: do you think the low-positive SYBR PCR results reflected the remnant non-infectious viral RNA from the challenge, or it was low-level of virus replication, or it was false-positive result? A short discussion on this would be very interesting.
We have rectified this in the revised version